# Probabilistic Surrogate Networks for Simulators with Unbounded Randomness

**Andreas Munk**[1]     **Berend Zwartsenberg**[1]     **Adam Ścibior**[2,1]     **Atılım Güneş Baydin**[3]     **Andrew Stewart**[4]

**Goran Fernlund**[4]          **Anoush Poursartip**[4,5]          **Frank Wood**[2,6,1]

[1]Department of Computer Science, University of British Columbia     [2]Inverted AI Ltd.
[3]Department of Engineering Science, University of Oxford
[4]Convergent Manufacturing Technologies Inc.
[5]Composites Research Network, University of British Columbia     [6]Mila, CIFAR AI Chair

## Abstract

We present a framework for automatically structuring and training fast, approximate, deep neural surrogates of stochastic simulators. Unlike traditional approaches to surrogate modeling, our surrogates retain the interpretable structure and control flow of the reference simulator. Our surrogates target stochastic simulators where the number of random variables itself can be stochastic and potentially unbounded. Our framework further enables an automatic replacement of the reference simulator with the surrogate when undertaking amortized inference. The fidelity and speed of our surrogates allow for both faster stochastic simulation and accurate and substantially faster posterior inference. Using an illustrative yet non-trivial example we show our surrogates' ability to accurately model a probabilistic program with an unbounded number of random variables. We then proceed with an example that shows our surrogates are able to accurately model a complex structure like an unbounded stack in a program synthesis example. We further demonstrate how our surrogate modeling technique makes amortized inference in complex black-box simulators an order of magnitude faster. Specifically, we do simulator-based materials quality testing, inferring safety-critical latent internal temperature profiles of composite materials undergoing curing.

## 1 INTRODUCTION

Stochastic simulators are accurate generative models that encode the relationship between random variables. Simulators can be used to reason about the relationship between latent variables and real world observations which the simulator is assumed to accurately model. Whether in aeronautical engineering [Wu et al., 2018], nonlinear flow physics [Veldman et al., 2007], finance [Raberto et al., 2001], or modeling the brain's blood flow [Perdikaris et al., 2016], simulators play an important role in design, diagnosis, and manufacturing. Unfortunately, complex simulators are often computationally expensive, ruling them out for just-in-time uses. This problem is exacerbated in stochastic simulators as these often need to be run many times to accurately estimate quantities of interest. A natural solution to this problem, known as surrogate modeling, is to construct a fast approximation to the reference simulator. This approach has found success in applications in various fields including computational fluid dynamics (CFD) [Glaz et al., 2010], aerospace engineering [Jeong et al., 2005], material science [Rikards et al., 2004] and quantum chemistry [Gilmer et al., 2017]. These surrogates learn to approximate the input to output mapping represented by a simulator, usually by fitting a regressor to samples drawn from the simulator. However, when such a simulator is stochastic, and especially when the number of internal random variables is unbounded, it is not immediately clear how to extend these ideas. It becomes impossible to write down a predefined parametrization. This is exactly the issue our work addresses: to provide a framework for surrogate modeling in the case where the number of random variables is generally unbounded.

Stochastic simulators can come in the form of (I) deterministic simulators with a fixed-dimensional vector of randomly distributed inputs (equivalent to a push-forward or structural equation model) or (II) a program that uses random variables internally. Constructing a surrogate for simulators that consumes randomness internally (type II) is done in one of two ways: (1) All internally utilized random variables are externalized and specified a priori as inputs, effectively transforming a type II stochastic simulator into type I. (2) Internal random variables are implicitly marginalized over, in which case any structural information internal to the simulators will be lost. In particular, considering (1), the randomness of a stochastic simulator can be abstracted to a single random number seed. Alternatively, and less extreme, samples of all

*Accepted for the 38th Conference on Uncertainty in Artificial Intelligence* (UAI 2022).

random variable types can be obtained by deterministically transforming $\mathcal{U}(0,1)$ pseudorandom numbers. So we can in theory transform a stochastic simulator of type II into a stochastic simulator of type I with $\mathcal{U}(0,1)$ distributed inputs. However, identifying all the internal random variables is in general impossible when a Turing complete language is used to specify the type II stochastic simulator that uses looping, branching, and other control flow constructs. This is because one must be able to identify or "address" all the random variables in advance. This is infeasible as the space of random variables can be countably infinite. So, any generic scheme to externalize the random variables of a type II stochastic simulator will involve *under-approximating* the original stochastic simulator, as only a finite number of variables can be externalized. Further like for option (2), such externalization discards structural information about the relationship between the otherwise internal random variables. As it is known that utilizing information about the structure of the model enables it to generalize better and lead to lower estimation loss [Bishop, 2006], it is desirable to retain all such information.

To address this we introduce a novel approach to surrogate modeling which captures and fully utilizes the structure of the stochastic simulator. Our surrogates, which we call *probabilistic surrogate networks* (PSNs), do not suffer from being an under-approximation. However, as we will discuss, it might be desirable to choose to execute them as under-approximations for practical purposes. Particularly, by framing the surrogate modeling problem in the context of probabilistic programming, our model architecture automatically replicates distributions over traces for a given reference simulator. This enables PSNs to generate interpretable sequences of latent variables that are fully compatible with the reference simulator. We achieve this by simultaneously learning to approximate the latent probability distributions *and* the control flow of the original simulator. Our method therefore targets simulators of type II in addition to type I, and we emphasize that it can handle simulators with arbitrarily many random variables. As a corollary we introduce a novel method for parameterizing a classifier defined over an unbounded number of classes.

Faster simulation via surrogate modeling is in itself useful. However, the speedup PSNs provide arguably has even greater impact on the "inversion" of simulators. Here inverting a simulator means performing Bayesian inference over latent variables given observed values of outputs. This definition blurs the line between stochastic simulators and probabilistic models and should be considered a key point of probabilistic programming [Baydin et al., 2019]. In this paper we illustrate how PSNs leverage faster inference by employing them in conjunction with the neural network based *inference compilation* (IC) framework [Le et al., 2017] and its PyProb [Baydin and Le, 2018] realization. PyProb is a probabilistic programming language (PPL) that enables

Bayesian inference in stochastic simulators written in other programming languages [Baydin et al., 2018], by intercepting and controlling random number draws during simulator execution. This process is explained elsewhere in full technical detail [van de Meent et al., 2018, chapt. 6]. PyProb was chosen due to several desirable features, such as automatic address construction.

## 2 BACKGROUND

### 2.1 PROBABILISTIC PROGRAMMING

The probabilistic programming paradigm equates a generative model with a program written in a probabilistic programming language (PPL). An inference backend takes the program and observed data and generates inference results, usually in the form of samples from a posterior distribution. PPLs can be broadly categorized as restricted, which limit the set of expressible models to ensure that particular inference algorithms can be applied [Lunn et al., 2009, Minka et al., 2018, Milch et al., 2005, Carpenter et al., 2017, Tran et al., 2016], and unrestricted (universal), which allow arbitrary models [Goodman et al., 2008, Mansinghka et al., 2014, Wood et al., 2014, Pfeffer, 2009, Goodman and Stuhlmüller, 2014, Bingham et al., 2018]. For instance, universal PPLs allow programs to contain for-loops where the number of iterations itself is stochastic and unbounded. For our purposes it is particularly important to note that extending an existing Turing-complete programming language with operations for sampling and conditioning results in a universal PPL [Goodman and Stuhlmüller, 2014]. For this reason existing stochastic simulators written in Turing-complete languages are programs in a universal PPL. As PSNs target universal PPLs we focus our discussion here on those.

A crucial concept is that of a *trace* of a probabilistic program. A trace is a sequence of random variables $(x_{a_t}, a_t)$ for $t = 1, \ldots, T$, where $a_t \in \mathcal{A}$ is an address [Wingate et al., 2011] of a random variable and $x_{a_t}$ its value. $\mathcal{A} = \{\alpha_1, \alpha_2, \ldots\}$ is a countable (potentially infinite) set of possible address values, which uniquely identify all random variables the simulator could ever produce. The purpose of the addresses is to identify the same random variables across different execution traces to facilitate correct inference. The trace length $T$ can vary between different executions of the same program and is generally unbounded.

Every probabilistic program specifies a joint distribution over the space of traces. Defining $\boldsymbol{x} = (x_{a_1}, \ldots, x_{a_T})$ and $\boldsymbol{a} = (a_1, \ldots, a_T)$ this distribution is denoted

$$p(\boldsymbol{x}, \boldsymbol{a}) = \prod_{t=1}^{T} p(a_t | x_{<a_t}, a_{<t}) p(x_{a_t} | x_{<a_t}, a_{\leq t}), \quad (1)$$

where $x_{<a_t} = \{x_{a_{t'}} | x_{a_{t'}} \in \boldsymbol{x}, t' < t\}$, $a_{<t} = a_{0:t-1}$,

$a_{\leq t} = a_{0:t}$, $a_0$ being the `begin-execution` address and $x_{<a_1} = \emptyset$. For each $t$, $p(a_t|x_{<a_t}, a_{<t})$ is the address transition probability distribution and $p(x_{a_t}|x_{<a_t}, a_{<t})$ is the distribution passed to the `sample` or `observe` statements in the program. It is these statements which allow for automatic inference in PPLs. The subset of $\boldsymbol{x}$ specified by `observe` statements is denoted $\boldsymbol{x}_{\text{obs}}$, while the remaining variables are denoted $\boldsymbol{x}_{\text{lat}}$ – i.e. those specified using `sample` statements. The goal of inference is to compute the posterior distribution $p(\boldsymbol{x}_{\text{lat}}|\boldsymbol{x}_{\text{obs}}) = \sum_{\boldsymbol{a}} p(\boldsymbol{x}_{\text{lat}}, \boldsymbol{a}|\boldsymbol{x}_{\text{obs}})$. It should be noted that in probabilistic programs $\boldsymbol{a}$ is always deterministic when conditioned on $\boldsymbol{x}$ making the marginalization of $\boldsymbol{a}$ trivial, as $p(a_t|x_{<a_t}, a_{<t}) = \delta(a_t - f(x_{<a_t}, a_{<t}))$, where $\delta(\cdot)$ is the Kronecker delta function and $f(\cdot)$ is the deterministic function defined by the simulator which specifies the address transition from address $a_t$ given $x_{<a_t}, a_{<t}$. However, modeling $\boldsymbol{a}$ as a random variable is essential to our PSN construction.

## 2.2 INFERENCE COMPILATION

Inference compilation (IC) [Le et al., 2017] is an amortized algorithm for performing inference in probabilistic programs using sequential importance sampling (SIS). It works by constructing an *inference network*, which constructs proposal distributions for all the latent random variables in the program, conditioned on the observed variables.

IC is essentially a self-normalizing importance sampler, specifically developed as an inference engine for probabilistic programming languages. IC infers $p(\boldsymbol{x}_{\text{lat}}|\boldsymbol{x}_{\text{obs}})$ using a proposal distribution $q(\boldsymbol{x}_{\text{lat}}|\boldsymbol{x}_{\text{obs}})$. It draws $K$ samples $\boldsymbol{x}_{\text{lat}}^k \overset{i.i.d.}{\sim} q$, computes the weights $w^k = \frac{p(\boldsymbol{x}_{\text{lat}}^k, \boldsymbol{x}_{\text{obs}})}{q(\boldsymbol{x}_{\text{lat}}^k|\boldsymbol{x}_{\text{obs}})}$, and approximates $p(\boldsymbol{x}_{\text{lat}}|\boldsymbol{x}_{\text{obs}}) \approx \sum_{k=1}^{K} w^k \delta(\boldsymbol{x}_{\text{lat}}^k - \boldsymbol{x}_{\text{lat}}) / \sum_{k=1}^{K} w^k$.

The proposal distribution $q$ factorizes in $t$ just like $p$. Subsequent conditional distributions in $q$ are constructed using a recurrent deep neural network, called the inference network. Specifically,

$$q_\phi(\boldsymbol{x}_{\text{lat}}, \boldsymbol{a}|\boldsymbol{x}_{\text{obs}}) = \prod_{x_{a_t}^{\text{lat}} \in \boldsymbol{x}_{\text{lat}}} q(x_{a_t}^{\text{lat}}|\eta_{a_t}(x_{<a_t}^{\text{lat}}, a_{<t}, \boldsymbol{x}_{\text{obs}}, \phi))$$
$$\times \prod_{t=1}^{T} q(a_t|x_{<a_t}, a_{<t}), \qquad (2)$$

where $x_{<a_t}^{\text{lat}} = \{x_{a_{t'}}|x_{a_{t'}} \in \boldsymbol{x}_{\text{lat}}, t' < t\}$, $\phi$ are the parameters of the inference network, and $\eta_{a_t}(\cdot)$ is the function computed by the neural network. We emphasize here how we explicitly write the address transitions as part of the inference problem, but note that in IC (and other similar inference engines) the address transitions in the posterior are defined as $q(a_t|x_{<a_t}, a_{<t}) \equiv p(a_t|x_{<a_t}, a_{<t})$.

The proposal $q_\phi$ is trained to match the true posterior $p(\boldsymbol{x}_{\text{lat}}, \boldsymbol{a}|\boldsymbol{x}_{\text{obs}}) \propto p(\boldsymbol{x}, \boldsymbol{a})$, where the distance between

the posteriors $p$ and $q_\phi$ is measured using the Kullback–Leibler (KL) divergence KL$(p \,||\, q_\phi)$. In order to match $q_\phi$ for all possible $\boldsymbol{x}_{\text{obs}}$ the expected KL divergence under the marginal $p(\boldsymbol{x}_{\text{obs}})$ is minimized.

It should be emphasized here, that for inference engines where $q(a_t|x_{<a_t}, a_{<t}) \equiv p(a_t|x_{<a_t}, a_{<t})$, the program and inference engine must run *concurrently*. That is, the address transitions are provided by the program via sampling from the dirac distribution $p(a_t|x_{<a_t}, a_{<t}) = \delta(a_t - a_t')$, where $a_t' = f(x_{<a_t}, a_{<t})$ is the deterministic address given $x_{<a_t}, a_{<t}$. This has two implications: (1) any surrogate modeling framework incorporated into a PPL framework must be able to provide such address transitions. (2) The runtime of inference engines relying on executing the reference simulator, like IC, will be computationally constrained by the computational complexity of the reference simulator. Such cases would be examples where surrogate models, like PSNs, in PPLs can drastically speed up the inference procedure.

## 3 PROBABILISTIC SURROGATE NETWORKS

PSNs are constructed to model a distribution over the trace space. They will replace the original program, thereby facilitating faster simulation and inference, provided the PSN is faster than the original program. PSNs factorize identically to the distribution of the original program specified in Eq. (1). Specifically, the distribution represented by a PSN is defined as

$$s_\theta(\boldsymbol{x}, \boldsymbol{a}) = \prod_{t=1}^{T} s(x_{a_t}|\xi_{a_t}(x_{<a_t}, a_{\leq t}; \theta)) \qquad (3)$$
$$\times s(a_t|\zeta_{a_{t-1}}(x_{<a_t}, a_{<t}; \theta)), \qquad (4)$$

where $\xi_{a_t}(\cdot)$ and $\zeta_{a_{t-1}}(\cdot)$ are neural networks. At the center of our PSNs, there is a recurrent neural network (RNN) that enables the density of $x_{a_t}$ to depend on all $x_{<a_t}$ and $a_{\leq t}$ that preceded it. We use $\theta$ to denote all parameters in the PSN, but note that the factors in Eq. (4) typically only use a subset of these. The PSN is trained to be close to $p(\boldsymbol{x}, \boldsymbol{a})$ in terms of the KL-divergence,

$$\mathcal{L}(\theta) = \text{KL}(p(\boldsymbol{x}, \boldsymbol{a}) \,||\, s_\theta(\boldsymbol{x}, \boldsymbol{a}))$$
$$= -\mathbb{E}_{p(\boldsymbol{x}, \boldsymbol{a})}[\log s_\theta(\boldsymbol{x}, \boldsymbol{a})] + \text{const}, \qquad (5)$$

$\mathcal{L}(\theta)$ is minimized by stochastic gradient descent, which requires calculating the unbiased gradient estimator

$$\boldsymbol{\nabla}_\theta \mathcal{L}(\theta) \approx -\frac{1}{N} \sum_{n=1}^{N} \boldsymbol{\nabla}_\theta \log s_\theta(\boldsymbol{x}^n, \boldsymbol{a}^n), \qquad (6)$$

with $(\boldsymbol{x}^n, \boldsymbol{a}^n) \overset{\text{iid}}{\sim} p(\boldsymbol{x}, \boldsymbol{a})$. It is crucial to distinguish between sampling repeatedly from an empirical distribution

$\hat{p}(\boldsymbol{x}, \boldsymbol{a}) \approx p(\boldsymbol{x}, \boldsymbol{a})$ (i.e. using a dataset) or sampling repeatedly from $p(\boldsymbol{x}, \boldsymbol{a})$ (online training) when calculating the gradient Eq. (6). Either approach puts different requirements on how $s_\theta(\boldsymbol{x}, \boldsymbol{a})$ can be constructed. In the former case, a naive but straightforward approach is to construct the possible address distributions and transitions by enumerating all traces in the dataset. However, this results in a surrogate model which is an under-approximation by construction. Furthermore, if new data containing unseen traces is later added to the dataset the surrogate must be reconstructed and retrained, which is computationally wasteful. Online training, on the other hand, requires the surrogate to grow dynamically as new data is sampled from the simulator. Our PSNs are designed to operate in the latter case and models the space of an unbounded number of random variables. Our method allows the PSN to grow dynamically as new traces are drawn from the simulator yet does not require the PSN to be retrained. This is due to PSNs having the property of being *measure preserving* with respect to a set of events $\mathcal{B}$ of particular interest as defined in Definition 1.

**Definition 1.** *Consider a probability space $(\Omega, \mathcal{F}, \mathbb{P}^g)$ and let $g \in \mathcal{G}$ be a function parameterizing the probability measure $\mathbb{P}^g$. Let $h : \mathcal{G} \to \mathcal{G}$ be a functional mapping such that $\mathbb{P}^{h(g)}$ is a probability measure associated with the probability space $(\Omega, \mathcal{F}, \mathbb{P}^{h(g)})$. Let $\mathcal{B} \subseteq \mathcal{F}$ be a set of subsets. We then say that $h$ is measure preserving with respect to $\mathcal{B}$ if,*

$$\mathbb{P}^{h(g)}(E) = \mathbb{P}^g(E), \forall E \in \mathcal{B}$$

We will show how to choose $\mathcal{B}$ such that the PSN grows to model newly encountered address transitions, while leaving the probability mass placed on known address transitions invariant to this expansion. Our method is inherently designed to work in the online setting but may also be used with an empirical trace distribution. The key to our method lies in how we model the probability measure associated with the set of infinitely many possible address transitions $a_{t+1}$ from address $a_t$. We accomplish this by parameterizing the probability measure in a way that dynamically allows "breaking" the probability measure into smaller pieces. Specifically, we consider the probability space $(\Omega_{a_t}, \mathcal{F}_{a_t}, \mathbb{P}^\zeta_{a_t})$, where $\Omega_{a_t}$ is the set of possible addresses the program can transition to from address $a_t$, $\mathcal{F}_{a_t}$ the $\sigma$-algebra, and $\mathbb{P}^\zeta_{a_t}$ the probability measure parameterized by a neural network $\zeta_{a_t}(x_{\le a_t}, a_{\le t}; \theta)$. We can without loss of generality partition $\Omega_{a_t}$ into transitions we are certain exist, $\mathcal{C}_{a_t}$, and transitions we are *uncertain* about, $\mathcal{U}_{a_t}$. We have that $\Omega_{a_t} = \mathcal{C}_{a_t} \cup \mathcal{U}_{a_t}$ and $\mathcal{C}_{a_t} \cap \mathcal{U}_{a_t} = \emptyset$. In practice $\mathcal{C}_{a_t}$ contains transitions observed during training and grows as we train the surrogate, while $\mathcal{U}_{a_t}$ contains transitions not yet encountered. We denote the size of known address transitions as $C = |\mathcal{C}_{a_t}|$, and define the neural network as a mapping $\zeta_{a_t} : \mathcal{A}_{<a_t} \times \mathcal{X}_{<a_t} \to \mathbb{R}^{C+1}$, where $a_{\le t} \in \mathcal{A}_{\le a_t}$ and $x_{\le a_t} \in \mathcal{X}_{\le a_t}$. From this we finally define

the parameterized probability measure,

$$\mathbb{P}^\zeta_{a_t}(E) = \frac{1}{Z} \begin{cases} e^{\zeta_{\gamma(c)}}, & \text{if } E = \{c\} \text{ and } c \in \mathcal{C}_{a_t} \\ e^{\zeta_{C+1}} & \text{if } E = \mathcal{U}_{a_t}, \end{cases} \quad (7)$$

where $\gamma : \mathcal{C}_{a_t} \to \{1, \ldots, C\}$ is a mapping from observed addresses to a unique "address index", $\zeta_i$ is the $i$th output of $\zeta_{a_t}(x_{\le a_t}, a_{\le t}; \theta)$, and $Z = e^{\zeta_{C+1}} + \sum_{c \in \mathcal{C}_{a_t}} e^{\zeta_{\gamma(c)}}$ is the normalization constant. Looking at Eq. (7), we see that the probability measure can be modeled using $\zeta_{a_t}$ in conjunction with the `softmax` function, $\mathbb{P}^\zeta_{a_t} = \text{softmax}(\zeta_{a_t}(x_{\le a_t}, a_{\le t}; \theta))$.

By modeling $\mathbb{P}^\zeta_{a_t}$ according to Eq. (7) we can consider the model to be a classifier which assigns probability to each address transition we know exists, while also assigning probability to yet unseen transitions. In order to relate Eq. (7) to the address transitions defined in Eq. (4) we note that $\forall c \in \mathcal{C}_{a_t}$ we can define $s(a_{t+1} = c|\zeta_{a_t}(x_{\le a_t}, a_{\le t}; \theta)) = \mathbb{P}^\zeta_{a_t}(\{c\})$. In a similar fashion we can $\forall u \in \mathcal{U}_{a_t}$ implicitly define $s(a_t = u|\zeta_{a_t}(x_{\le a_t}, a_{\le t}; \theta))$ in terms of $\mathbb{P}^\zeta_{a_t}(\mathcal{U}_{a_t}) = \sum_{u \in \mathcal{U}_{a_t}} \mathbb{P}^\zeta_{a_t}(\{u\})$, where the summation is justified as the set of all addresses (and therefore $\mathcal{U}_{a_t}$) is countably infinite. The address transition probability, $s(a_t = u|\zeta_{a_t}(x_{\le a_t}, a_{\le t}; \theta))$, would be one of the terms in the sum. To provide some intuition on how to use the parameterized probability measure in Eq. (7), we now describe how our PSNs grows during the optimization procedure. For every set of samples of size $N$ used to calculate the gradient estimator (i.e. a mini-batch), enumerate all the addresses and their transitions. For each address consider all new address transitions, which are transitions *not* found in $\mathcal{C}_{a_t}$. Let the set of newly encountered address transitions be denoted $\mathcal{K}_{a_t}$ and its size be denoted $K = |\mathcal{K}_{a_t}| \le N$. We then expand the neural network $\zeta_{a_t}$ and refer to the expansion as $\tilde{\zeta}_{a_t} : \mathcal{A}_{<a_t} \times \mathcal{X}_{<a_t} \to \mathbb{R}^{C+K+1}$. The expansion, $\tilde{\zeta}_{a_t}$, has its own learnable parameters that are derived directly from $\zeta$ and parameterizes a new probability measure $\mathbb{P}^{\tilde{\zeta}}_{a_t}$. We carry out the expansion so that $\mathbb{P}^{\tilde{\zeta}}_{a_t}$ is given by,

$$\mathbb{P}^{\tilde{\zeta}}_{a_t}(E) = \frac{1}{\tilde{Z}} \begin{cases} e^{\tilde{\zeta}_{\tilde{\gamma}(c)}}, & \text{if } E = \{c\} \text{ and } c \in \mathcal{C}_{a_t} \cup \mathcal{K}_{a_t} \\ e^{\tilde{\zeta}_{C+K+1}}, & \text{if } E = \tilde{\mathcal{U}}_{a_t} \end{cases} \quad (8)$$

where $\tilde{\mathcal{U}}_{a_t} = \mathcal{U}_{a_t} \setminus \mathcal{K}_{a_t}$, $\tilde{Z}$ the new normalization constant and $\tilde{\gamma} : \mathcal{C}_{a_t} \times \mathcal{K}_{a_t} \to \{1, \ldots, C, C+1, \cdots, C+K\}$ the new index mapping, which is equal to $\gamma$ for the same addresses already in $\mathcal{C}_{a_t}$. Specifically we have $e^{\tilde{\zeta}_{\tilde{\gamma}(c)}} = e^{\zeta_{\gamma(c)}}$ if $c \in \mathcal{C}_{a_t}$, $e^{\tilde{\zeta}_{\tilde{\gamma}(c)}} = e^{\zeta_{C+1} - \log(K+1)}$ if $c \in \mathcal{K}_{a_t}$, and $e^{\tilde{\zeta}_{C+K+1}} = e^{\zeta_{C+1} - \log(K+1)}$. This choice, Eq. (8), leads to the following theorem, which we prove in Appendix A.1,

**Theorem 1.** *Consider a probability measure $\mathbb{P}^\zeta_{a_t}$ characterized by a neural network $\zeta_{a_t} \in \mathcal{G}$ according to Eq. (7). Consider also a sample of traces of size $N$ which for each address $a_t$ contain a set $\mathcal{K}_{a_t}$ of new address transitions.*

*Let the expansion procedure represented by Eq. (8) be defined as the function $h : \mathcal{G} \to \mathcal{G}$ such that $\tilde{\zeta}_{a_t} = h(\zeta_{a_t})$. If $\mathcal{B} = 2^{\mathcal{C}_{a_t}} \cup \{\mathcal{U}_{a_t}\} \subseteq \mathcal{F}_{a_t}$ where $2^{\mathcal{C}_{a_t}}$ denotes the powerset of $\mathcal{C}_{a_t}$, then for all addresses $a_t$, the functional mapping $h$ is measure preserving with respect to $\mathcal{B}$ as defined in Definition 1, and*

$$\mathbb{P}_{a_t}^{\tilde{\zeta}}(E) = \mathbb{P}_{a_t}^{\zeta}(E), \forall E \in \mathcal{B}$$

Once a new probability measure is created, the new transitions found in $\mathcal{K}_{a_t}$ are added to $\mathcal{C}_{a_t}$. In Fig. 1 we show an illustration of the expansion process and we provide further details on how to expand the PSNs when encountering new address transitions in Appendix B where we also provide Algorithm 1. Following the expansion of the PSN, the update of the PSN parameters, $\theta$, is carried out by calculating the gradient estimator and performing gradient descent. This procedure is repeated until convergence. Additional details and design choices of PSNs can be found in Appendix C.

## 3.1 EVALUATING AND EXECUTING PSNS

The construction of our PSNs described above ensures that the surrogate models define a probability measure on spaces with an unbounded number of random variables. In particular, and we prove this in Appendix A.2,

**Theorem 2.** *Let $s(\boldsymbol{x}, \boldsymbol{a})$ be a surrogate model using PSNs. Then any trace $(\boldsymbol{x}, \boldsymbol{a}) \sim p(\boldsymbol{x}, \boldsymbol{a})$ can be evaluated under $s(\boldsymbol{x}, \boldsymbol{a})$.*

While Theorem 2 guarantees evaluation for all possible traces generated by the reference simulator, the surrogate $s$ is only likely to provide accurate density estimates for traces for which all addresses have been encountered during training. As such, at evaluation time when training is complete, it is of more practical use to place zero probability measure on traces containing unknown addresses. The justification of this choice becomes more apparent when discussing the execution of PSN-based surrogate models. Such executions start with the `begin-execution` address, after which the surrogate samples a new address from the transition distribution, a value is sampled from the distribution at the sampled address, after which a new address transition is sampled, *etcetera*, finishing only when the surrogate samples an `end-execution` address. The procedure is illustrated in more detail in Fig. 1 in Appendix C. The question now arises what should happen if the surrogate samples an `unknown` address at any point during its execution. Recall that at each address $a_t$ the probability associated with such an event is $\mathbb{P}_{a_t}^{\zeta}(\mathcal{U}_{a_t})$. One straightforward approach would be to (1) Generate a new arbitrary address including the possibility to generate an `end-execution` address. (2) If the new address is not an `end-execution` address then expand the PSN according to Eq. (8) in order to accommodate the

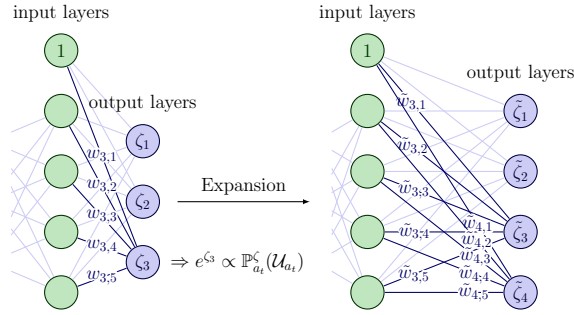

Figure 1: Illustration of the PSN expansion relating Eq. (7) and Eq. (8). The expansion takes place in the final address transition prediction layer, and the new weights $\tilde{w}_{\cdot,\cdot}$ relate to the former weights $w_{\cdot,\cdot}$ as follows: (1) for the weights associated with known address transitions we have $\tilde{w}_{i,j} = w_{i,j}$ for all $i, j \in \{1, 2\} \times \{1, \ldots, 5\}$. (2) For the weights associated with the unknown addresses and newly encountered addresses we have for all $i \in \{1, \ldots, 5\}$ that $\tilde{w}_{3,i} = \tilde{w}_{4,i}$ where $\tilde{w}_{3,i} = w_{3,i} - \log(K + 1)$. In this case, with one newly encountered address, we have $K = 1$.

newly generated address. (3) Sample some distribution from a prior distribution over distributions. (4) Repeat until an `end-execution` address is generated. Clearly, the produced traces from such a procedure will almost certainly have zero probability under the reference simulator, and would yield spurious results. To remedy this, we instead decide to only allow transitions between addresses encountered during training. Specifically, whenever an `unknown` address is sampled, we keep resampling until a known address is sampled, leading to the following adjusted address transition probabilities for all $c \in \mathcal{C}_{a_t}$,

$$\begin{aligned} P(a_{t+1} = c | A) &= \frac{P(A|c) s(c|\zeta_{a_t}(x_{\le a_t}, a_{\le t}; \theta))}{P(A)} \\ &= \frac{s(c|\zeta_{a_t}(x_{\le a_t}, a_{\le t}; \theta))}{1 - \mathbb{P}_{a_t}^{\zeta}(\mathcal{U}_{a_t})}, \end{aligned} \quad (9)$$

where $A$ denotes the event that we accept the address transition $a_t \to a_{t+1}$.

## 3.2 PRACTICAL LIMITATIONS

While PSNs target programs written in universal PPLs, there are practical considerations accompanying (1) the rejection sampling step of address transitions and (2) the proposed use of RNNs as the core of the PSN. Regarding (1), the rejection sampling step is equivalent to placing zero probability mass on traces that were not observed during training. In general this results in the adjusted address transition probabilities, Eq. (9), to become slightly biased. Additionally, this implies that PSNs become an under-approximation of the target simulator, which may have non-zero probability on certain traces where the PSN places no probability mass. In

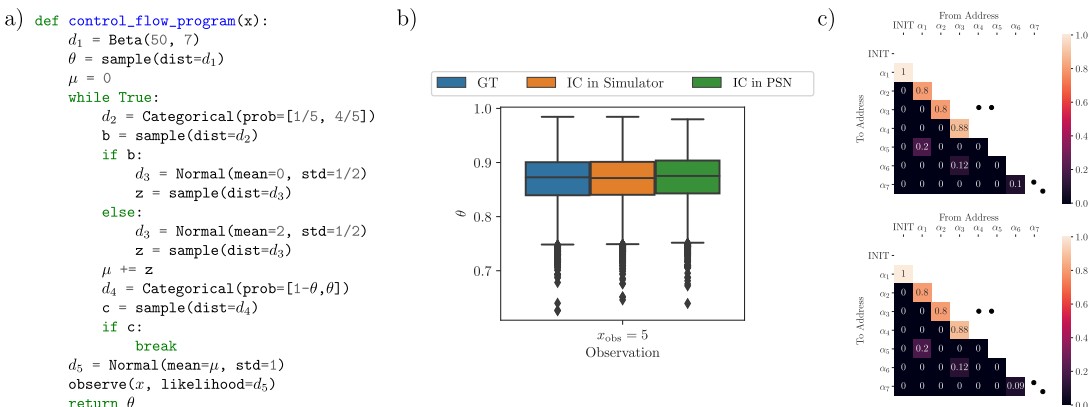

Figure 2: (a) Program containing stochastic control flow in the form of a for-loop with a nested if-else statement. The task here is to perform posterior inference about $\theta$ given the observed value of $x$. (b) Each boxplot represents the estimated posterior distribution of $\theta$ conditioned on $x = 5$. We see that inference using IC in the simulator and using IC in the PSN both are identical to GT. (c) shows a subset of the address transitions with high probability for the original program (top) and our PSN (bottom). We observe identical transition probabilities which, together with (b), shows that the PSN is able to approximate models with complex control flow.

the limit of observing all possible address transitions these issues simply vanish, while practically the more traces are observed, the less likely it becomes that important addresses with high probability mass are missed. Concerning (2), the choice of using RNNs to model the flow of information (i.e. inter-variable dependencies) was made as it has proven very effective in practice. Notwithstanding, while RNNs are capable, in theory, of emulating Turing machines [Weiss et al., 2018, Siegelmann, 1998, Siegelmann and Sontag, 1994, 1995, Chen et al., 2018], finite memory and floating point precision make them finite state machines in practice. For target programs, which require storing information on e.g. a potentially infinitely growing stack, we would not, in general, expect RNNs to model said programs arbitrarily well. This does not, however, influence the results in Theorems 1 and 2 which are agnostic to the specific implementation of the dependency model. Rather, it implies that the size of the RNN needs to be chosen appropriately to ensure that accurate surrogates are learned. Put in different words, the approximating distribution has limited flexibility, but its support is guaranteed to be correct by Theorem 2. If RNNs turn out insufficient, we suggest considering differentiable neural computers [Graves et al., 2016] as a potential suitable alternative, as it has access to external memory. Furthermore, previous work by Harvey et al. [2019] suggests that the transformer architecture [Vaswani et al., 2017] might also be a good alternative choice in some cases.

### 3.3 COMPLEXITY ANALYSIS

We limit the complexity analysis to pertain to the number of addresses encountered during training - a set we denote $A$. We start by considering the worst-case scenario, where the possible addresses transitions of some program is as follows:

Order all addresses on a single line in the order in which they could appear. If a transition can occur from any address to any other address following it, the computational complexity must be $\mathcal{O}(A^2)$. This is also true memory-wise. This is because from any particular address we must calculate the transition probability to any of the addresses following it. This includes storing model parameters to each of those potential addresses.

How the PSNs compare to the reference simulator complexity-wise cannot generally be determined. We imagine that the reference simulator in many cases has similar complexity. For instance, if the various address transitions are due to if-elif-else statements in a program, the reference simulator may calculate all logical clauses leading to complexity $\mathcal{O}(A^2)$. However, there might exist an equivalent program which is much more efficient in how it determines its state transitions, possibly even $\mathcal{O}(A)$, but we cannot in general make such guarantees. Similarly, there may exist other programs which scale much worse, say $\mathcal{O}(x^A)$, $x > 1$ - we can imagine programs which do complex computations that reason about all possible future and past states.

Ultimately, what matters in determining whether or not to use a PSN to replace the reference simulator is the wall-clock time of the PSN versus the reference simulator.

## 4 EXPERIMENTS

### 4.1 STOCHASTIC CONTROL FLOW

Here we present an experiment that highlights the PSN's capability to learn a model's address transitions. Fig. 2(a)

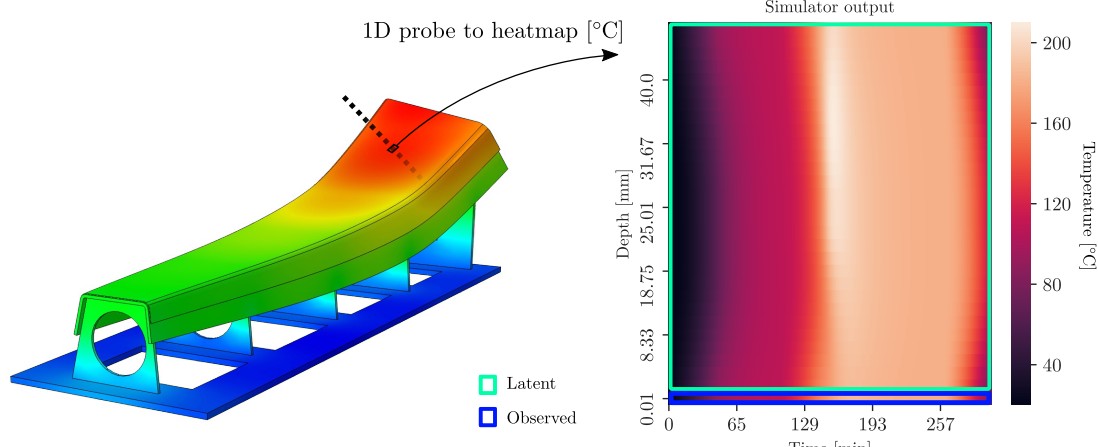

Figure 3: Composite manufacturing involves an uncured composite material being laid up onto a tool, of known material, which are then placed in an autoclave where a predefined pressure and heating cycle is imposed (left). We consider a 1D simulation of this process as a function of time, leading to the 2D heatmap (right). The set of latent variables are heat transfer coefficients and thicknesses and internal temperature (green box). The observed variables are temperature configuration of the autoclave, air temperatures, and tool temperature (blue box, measured at the bottom surface of the tool).

shows a program with complex stochastic control flow, where the aim is to perform posterior inference of $\theta$ given the observed value of $x = 5$ using a trained PSN. Fig. 2(b) shows boxplots representing the estimated posterior distribution of $\theta$ conditioned on $x = 5$. The inference results are obtained using either MCMC, specifically Lightweight Metropolis-Hastings (LMH) [Wingate et al., 2011], with a chain length of 1,000,000 samples (denoted GT for ground truth) or IC in either the simulator or PSN using 10,000 resampled importance weighted samples. To evaluate the address transition capability we look at Fig. 2(c), which shows a subset of address transitions with high probability observed across 50,000 generated samples from the model (top) and PSN (bottom). We observe that the three posteriors and the address transition probabilities are identical. Together these results show that the PSN has successfully approximated the program including the address transitions associated with the original program. Further evidence can be found in Appendices D.1.1 and D.4.

## 4.2 PROGRAM SYNTHESIS

Next we consider the question of whether the machinery as it is presented here is able to capture relevant connections between the addresses as they are available. As touched upon in Section 3.2, RNNs are finite state machines in practice and so may be insufficient in accurately modeling the inter-variable dependencies. To shed light on this we provide an experiment that showcases that RNNs can, in practice, model programs which require access to dynamically growing memory. Particularly, we learn a surrogate for a model that generates valid Python programs. We use a subset of the Python syntax that allows `if`, `else`, and `for` state-

ments, to an unbounded nesting depth, corresponding to piecewise linear functions. Example programs and full technical details of the simulator can be found in Appendix D.5. The crucial element of this experiment is the existence of a stack in the original simulator, that tracks the opening and closing of conditionals, and determines at any time what constitutes a valid next line. The surrogate has to store this information in the RNN hidden state, or alternatively, learn the valid continuations that belong to a certain unbounded collection of addresses. We judge the quality of PSNs by the fraction of valid programs that are generated. As the validity of programs allows for direct evaluation without performing inference, we omit the latter. We find the percentage of valid programs to be 99.62% (50k samples). We thus conclude that in practice, the use of an RNN for our method is easily sufficient for a task requiring the simulation of a program stack.

## 4.3 PROCESS SIMULATION OF COMPOSITE MATERIALS

In this experiment we train a surrogate model for a commercial heat-transfer finite element analysis simulator, depicted in Fig. 3, that is used to model the cure cycle for composite aircraft (e.g. Boeing) parts. We show how to use inference to estimate the temperature of the part in regions that cannot be accessed non-invasively. Such results are critical for determining whether the part is safe or not. The particular simulator used is RAVEN which simulate the curing process of composite materials, a proprietary software developed by Convergent Manufacturing Technologies [2019]. RAVEN is used in the aerospace and automotive industries to evaluate key performance metrics for part manufactur-

a)

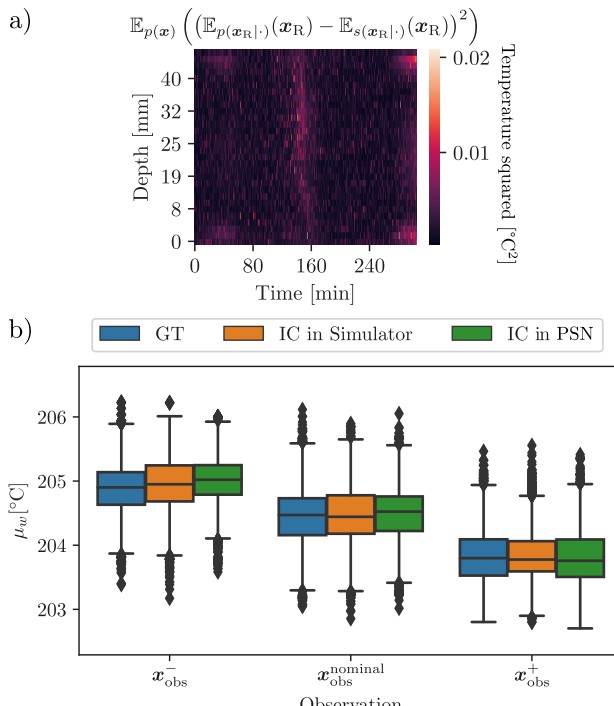

b)

Figure 4: (a) Shows the expected squared difference between the output ($\boldsymbol{x}_{\mathrm{R}} = \boldsymbol{x}_{\mathrm{RAVEN}}$) from the PSN and output from the simulator. We observe negligible errors throughout, with small peaks around time $160\,\mathrm{min}$ and towards the end of the heating process at the top/bottom. Each boxplot in (b) represents the estimated posterior distribution (conditioned on either $\boldsymbol{x}_{\mathrm{obs}}^{-}$, $\boldsymbol{x}_{\mathrm{obs}}^{\mathrm{nominal}}$, or $\boldsymbol{x}_{\mathrm{obs}}^{+}$) over a fixed time window $f(\boldsymbol{x}) = \mu_w$. We see that IC in the PSN yields effectively the same posterior as IC in the simulator across all observations. In all cases inference using IC agrees with the ground truth (GT).

ing design with the ultimate goal of decreasing manufacturing cost whilst retaining part performance and safety. Physical observations of the material's internal temperature during manufacturing are expensive, if not impossible, and manufacturers would prefer to infer the internal state of the material given less expensive external measurements. Fig. 3 illustrates this process and the experimental setup. Using probabilistic programming and our PSNs, we seek to infer the internal state of the material conditioned on realistically observable quantities. We evaluate the quality of the PSN by considering the expectation of $\boldsymbol{x}_{\mathrm{RAVEN}}$ (material temperature during processing) conditioned on the RAVEN configurations, $\boldsymbol{x}_{\mathrm{config}}$, under (1) the model distribution $\mathbb{E}_{p(\boldsymbol{x}_{\mathrm{RAVEN}}|\boldsymbol{x}_{\mathrm{config}})}[\boldsymbol{x}_{\mathrm{RAVEN}}]$ and (2) the PSN distribution $\mathbb{E}_{s(\boldsymbol{x}_{\mathrm{RAVEN}}|\boldsymbol{x}_{\mathrm{config}})}[\boldsymbol{x}_{\mathrm{RAVEN}}]$. Fig. 4(a) shows generally negligible expected squared errors between the surrogate and simulator outputs, and we provide additional results in Appendix D.3 that showcase the efficacy of the trained PSN. Small peaks are, however, observed in Fig. 4(a) at around time $t = 160\,\mathrm{min}$, as well as towards the end of

Table 1: We estimate $\hat{\mu}_w \approx \mathbb{E}_{p(\boldsymbol{x}_{\mathrm{lat}}|\boldsymbol{x}_{\mathrm{obs}})}[\mu_w]$ under the posterior using SIS denoted GT, IC in simulator (ICS) and IC in PSN (ICP) using 15,000 traces, and report the associated effective sample size (ESS). We provide six different estimates; three posteriors and three observations $\boldsymbol{x}_{\mathrm{obs}}^{-}$, $\boldsymbol{x}_{\mathrm{obs}}^{\mathrm{nominal}}$, and $\boldsymbol{x}_{\mathrm{obs}}^{+}$. We observe that the PSN estimates matches that of the GT and IC in the simulator.

|  | $\boldsymbol{x}_{\mathrm{obs}}^{-}$ | | $\boldsymbol{x}_{\mathrm{obs}}^{\mathrm{nominal}}$ | | $\boldsymbol{x}_{\mathrm{obs}}^{+}$ | |
|---|---|---|---|---|---|---|
|  | $\hat{\mu}_w$ | ESS | $\hat{\mu}_w$ | ESS | $\hat{\mu}_w$ | ESS |
| GT | 204.90 | 259 | 204.46 | 304 | 203.82 | 399 |
| ICS | 204.96 | 158 | 204.46 | 340 | 203.83 | 204 |
| ICP | 205.01 | 173 | 204.49 | 279 | 203.80 | 292 |

the heating process, which is where the internal temperature exhibits the most rapid changes.

To evaluate the quality of performing inference (using IC) in the PSN we consider the scenario where we only observe the configurations, air and surface temperatures of the curing process, $\boldsymbol{x}_{\mathrm{obs}}$. The latent variables $\boldsymbol{x}_{\mathrm{lat}}$ are the dimensions of the material, the heat transfer coefficients and the internal temperature during curing. We then consider the function $f(\boldsymbol{x}_{\mathrm{lat}}) = \mu_w$, being the empirical mean of the internal temperature of the material across the time window $w = [155\,\mathrm{min}, 165\,\mathrm{min}]$ (chosen to be close to peak temperatures) and at a fixed depth $30\,\mathrm{mm}$ (chosen to be somewhere near the upper quarter of the material). We then estimate $\hat{\mu}_w \approx \mathbb{E}_{p(\boldsymbol{x}_{\mathrm{lat}}|\boldsymbol{x}_{\mathrm{obs}})}[f(\boldsymbol{x}_{\mathrm{lat}})] = \mathbb{E}_{p(\boldsymbol{x}_{\mathrm{lat}}|\boldsymbol{x}_{\mathrm{obs}})}[\mu_w]$ using IC with the same inference network $q(\boldsymbol{x}_{\mathrm{lat}}|\boldsymbol{x}_{\mathrm{obs}})$ used for performing inference in both the surrogate and the model. As a ground truth posterior, we employ SIS where the proposal distribution is the prior $q(\boldsymbol{x}_{\mathrm{lat}}|\boldsymbol{x}_{\mathrm{obs}}) = p(\boldsymbol{x}_{\mathrm{lat}})$ and denote it GT. To evaluate the effect of amortized inference we consider conditioning on three different observations $\boldsymbol{x}_{\mathrm{obs}}^{-}$, $\boldsymbol{x}_{\mathrm{obs}}^{\mathrm{nominal}}$, and $\boldsymbol{x}_{\mathrm{obs}}^{+}$ each corresponding to an observation produced by the simulator with input values and temperature settings well below, equal to, and well above the nominal values respectively. In all cases inference is performed using 15,000 traces (SIS particles) and we summarize the results in Table 1. We show that performing inference in the PSN yields approximately the same results as inference in the simulator. We only find small deviations when observing $\boldsymbol{x}_{\mathrm{obs}}^{-}$ where the PSN seems to barely overestimate $\hat{\mu}_w$ compared to the GT. To get a sense of how our traces are distributed we show in Fig. 4(b) boxplots representing the posterior distribution from which we estimate $\mathbb{E}_{p(\boldsymbol{x}_{\mathrm{lat}}|\boldsymbol{x}_{\mathrm{obs}})}[\mu_w]$. Each boxplot is made by resampling the 15,000 importance weighted samples. The results confirm that inference in the PSN yields similar posteriors compared to inference in the simulator. These boxplots also illustrate why $\hat{\mu}_w$ was slightly overestimated when doing inference in the PSN; when observing $\boldsymbol{x}_{\mathrm{obs}}^{-}$ the posterior is shifted slightly upwards compared to the GT.

The advantage of using the PSN is that we maintain high accuracy in the posterior estimates with a speedup factor of 15.32 when comparing the number of traces generated per second. Furthermore, in cases where we simply seek to produce faster simulations (not for the sake of inference), the PSN provides an even greater speedup factor of 90.16. The additional speedup is due to dropping the overhead of performing inference. The exact running times and model specifications can be found in Appendices D.1.2 and D.2.

## 5 RELATED WORK

As far as the authors of this paper are aware, the PSN is the first framework for learning surrogate models that models simulators containing a potentially unbounded number of random variables by automatically extracting and using a simulator's latent structure. Surrogate modeling is, however, a topic that dates back several decades and is fundamentally a regression problem, where the surrogate predicts the output of the model for a given input. Currently, the most commonly used methods for constructing deterministic surrogate models [Razavi et al., 2012] include Kriging [Simpson et al., 2001, Sacks et al., 1989], support vector machines (SVMs) [Willcox and Megretski, 2005], radial basis functions (RBFs) [Hussain et al., 2002, Mullur and Messac, 2006], and neural networks (NNs) [Tompson et al., 2017, Khu and Werner, 2003, Gilmer et al., 2017], while methods like the stochastic Kriging [Hamdia et al., 2017] allow for stochastic surrogate modeling. Notwithstanding, such commonly used methods are incompatible with simulators with an unbounded number of variables.

Finally, the idea of learning trace executions using LSTMs has been studied before, see for example neural programmer-interpreters (NPI) [Reed and De Freitas, 2015]. Methods like NPIs are trained to predict the sequence of called subroutines used to solve specific tasks like sorting or image rotation. As such, NPIs make no attempt to abstract away the predicted subroutines. That is, if any subroutine causes a computational bottleneck, NPIs cannot decrease the computational cost. This is fundamentally different to our PSN surrogate method which aims to model the entire simulator.

## 6 CONCLUSIONS

We have proposed *probabilistic surrogate networks*, a novel approach to surrogate modeling that considers not only the distributions in stochastic simulators but the stochastic structure of the simulator itself. Our main contribution is to develop a construction in which the surrogates allow for the description of a dynamically growing number of random variables, while maintaining consistency of the assigned probability measure as new variables are encountered. Such a framework is a requirement for producing surrogates for arbitrary simulators potentially containing an unbounded number of random variables. Using a real-world process simulation of composite materials as an example, we have shown that our approach provides significant computational speedup in inference problems using inference compilation, while preserving the quality of inference results that are indistinguishable from the ground truth.

## Acknowledgements

We acknowledge the support of the Natural Sciences and Engineering Research Council of Canada (NSERC), the Canada CIFAR AI Chairs Program, and the Intel Parallel Computing Centers program. Additional support was provided by UBC's Composites Research Network (CRN), and Data Science Institute (DSI). This research was enabled in part by technical support and computational resources provided by WestGrid (www.westgrid.ca), Compute Canada (www.computecanada.ca), and Advanced Research Computing at the University of British Columbia (arc.ubc.ca).

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
