# OpenReview forum: "Probabilistic Surrogate Networks for Simulators with Unbounded Randomness"
_auai.org/UAI/2022/Conference — UAI 2022 Poster_

### Official Review · Reviewer_JjH3 · 2022-04-12

**Q2(1) Originality/Novelty:** 3
**Q2(2) Significance/Impact:** 3
**Q2(3) Correctness/Technical Quality:** 3
**Q2(6) Clarity Of Writing:** 3
**Q6 Overall Score:** 7
**Q8 Confidence In Your Score:** 2

**Q1 Summary And Contributions:**

In this paper, the authors focus on the challenging problem of computer simulations. In particular, they propose an approach based on deep neural networks of stochastic simulators.
In particular, they make contributions on the complex task of handling a dynamically evolving number of random variables.

**Q2 Assessment Of The Paper:**

More detailed information regarding each of these aspects is given below:

**Q2(4) Quality Of Experiments (Optional):**

2: Fair: The experimental evaluation is weak: important baselines are missing, or the results do not adequately support the main claims.

**Q2(5) Reproducibility:**

2: Fair: Key resources (e.g., proofs, code, data) are unavailable but key details (e.g., proof sketches, experimental setup) are sufficiently well-described for an expert to confidently reproduce the main results.

**Q3 Main Strengths:**

The paper presents a novel architecture that is shown to work well empirically.

Although the approach can be limited by the capabilities of the RNNs used, this limitation is present in any ML based simulator. The empirical evaluation shows however that they might be capable enough for some interesting set of applications.


**Q4 Main Weakness:**

Although the authors show a speed improvement between one and two orders of magnitude. It is not clear how the method scales with regards to accuracy, training and testing time.

**Q5 Detailed Comments To The Authors:**

The authors present a well-written paper on an interesting intersection of surrogate simulators, probabilistic programming and deep neural networks.

And I would say that the only issues that were not developed further are concerning the scaling limitations which might be a factor for practitioners.
There are of course natural questions regarding the limits of the RNNs too, and whether they can be overcome by more powerful architectures such as transformers in practical settings.

**Q7 Justification For Your Score:**

The paper presents a working approach as shown by the empirical evidence.

**Q9 Complying With Reviewing Instructions:**

1: Yes.

---

### Official Review · Reviewer_XDYV · 2022-04-13

**Q2(1) Originality/Novelty:** 3
**Q2(2) Significance/Impact:** 2
**Q2(3) Correctness/Technical Quality:** 3
**Q2(6) Clarity Of Writing:** 2
**Q6 Overall Score:** 6
**Q8 Confidence In Your Score:** 1

**Q1 Summary And Contributions:**

This paper looks at building surrogates for stochastic simulators where the number of random variables is itself unbounded. The goal is to replace the stochastic simulators with the surrogates, as they are to result in faster approximate inference.

**Q2 Assessment Of The Paper:**

More detailed information regarding each of these aspects is given below:

**Q2(4) Quality Of Experiments (Optional):**

3: Good: The experimental evaluation is adequate, and the results convincingly support the main claims.

**Q2(5) Reproducibility:**

4: Excellent: Key resources (e.g., proofs, code, data) are available and key details (e.g., proof sketches, experimental setup) are comprehensively described for competent researchers to confidently and easily reproduce the main results.

**Q3 Main Strengths:**

The main strength of this approach is that it can handle unbounded length of random variables, which seems to be the novelty here. The paper also has an interesting set of experiments, including a real-world application.

**Q4 Main Weakness:**

The presentation of the work can be improved for readers (especially new readers like me who are not well versed with this area).

**Q5 Detailed Comments To The Authors:**

- Can the authors highlight what does the authors mean by address transitions?

- Between sec 2.2 and sec 3, I am guessing q has been replaced by s. Can the authors clarify this?

- In sec 3, it says "In fact, our PSNs allow for new data to later be
added to the empirical distribution with the PSNs being
continually trained using the updated empirical distribution." How exactly would that work?

- In sec 3.1, it says "As such, at evaluation time when training is complete, it is of more practical use to place zero probability measure on traces containing unknown addresses." What if we add a small probability mass on traces containing unknown addresses?

- Based on reading the paper, I realised that what authors mean by the random variables are unbounded, is that the sequences of x and a can be arbitrarily long. However, a can have values in only a finite set. Is that correct?

**Q7 Justification For Your Score:**

In all honestly, I am not really familiar with this area, and my score is based on the strengths that I gathered from this paper. I'd have no problems if my score is completely discounted.

**Q9 Complying With Reviewing Instructions:**

1: Yes.

---

### Official Review · Reviewer_n7EN · 2022-04-18

**Q2(1) Originality/Novelty:** 3
**Q2(2) Significance/Impact:** 3
**Q2(3) Correctness/Technical Quality:** 3
**Q2(6) Clarity Of Writing:** 3
**Q6 Overall Score:** 7
**Q8 Confidence In Your Score:** 3

**Q1 Summary And Contributions:**

This paper proposes a framework for learning surrogate models as under-approximation for stochastic simulators. Specifically, it aims at simulators that would potentially contain an unbounded number of variables. In the proposed framework, the conditional distributions on trace space are parameterized by RNNs and they are trained via variational objectives. Empirical evaluations include stochastic control flow, Python program generation and simulator for real-world problems.

**Q2 Assessment Of The Paper:**

More detailed information regarding each of these aspects is given below:

**Q2(4) Quality Of Experiments (Optional):**

3: Good: The experimental evaluation is adequate, and the results convincingly support the main claims.

**Q2(5) Reproducibility:**

3: Good: Key resources (e.g., proofs, code, data) are available and key details (e.g., proofs, experimental setup) are sufficiently well-described for competent researchers to confidently reproduce the main results.

**Q3 Main Strengths:**

The problem of approximating simulators with a potentially unbounded number of variables is interesting since the unbounded randomness naturally arises in various simulators. This paper seems to provide a decent solution to this problem setting as shown in the empirical evaluations in both synthetic and real-world simulators. The paper is overall well-written and it is easy to follow.


**Q4 Main Weakness:**

Since the point of having surrogate models is to have a fast approximation to simulators that are expensive to run, I think the authors should present a complexity analysis of the proposed framework. In the proposed framework, it seems that computing the partition function in Eq (7) is expensive since it requires to go through all the transitions in the training set. Also, this normalization constant would be re-computed several times as shown in Eq (8). I wonder if this would greatly impede the efficiency of the proposed framework and what is the specific complexity for it.

**Q5 Detailed Comments To The Authors:**

- As mentioned in [Main Weakness], I wonder if the authors could provide some complexity analysis of the proposed framework.
- The specific structures/architectures of the RNNs used in the experiments are missing. I wonder what the sizes of these RNNs are that give decent performances.
- The boxplot in both Fig.1 and Fig. 3 have overlapping boxes, which authors might want to modify.
- The font size in Fig. 1 c) is too tiny to read.


**Q7 Justification For Your Score:**

The contributions in this work seem solid to me.

**Q9 Complying With Reviewing Instructions:**

1: Yes.

---

### Official Review · Reviewer_gTaf · 2022-04-19

**Q2(1) Originality/Novelty:** 3
**Q2(2) Significance/Impact:** 2
**Q2(3) Correctness/Technical Quality:** 3
**Q2(6) Clarity Of Writing:** 4
**Q6 Overall Score:** 7
**Q8 Confidence In Your Score:** 4

**Q1 Summary And Contributions:**

The goal of the paper is to enable the construction of probabilistic surrogate neural network models that correspond to probabilistic programs and retains its interpretable structure and control flow.  The framework allows stochastic, unbounded number of random variables, and enables amortized inference. The utility of the framework is illustrated in pedagogic examples as well as a safety critical application involving temperature profile estimation in composite materials.

**Q2 Assessment Of The Paper:**

More detailed information regarding each of these aspects is given below:

**Q2(4) Quality Of Experiments (Optional):**

3: Good: The experimental evaluation is adequate, and the results convincingly support the main claims.

**Q2(5) Reproducibility:**

3: Good: Key resources (e.g., proofs, code, data) are available and key details (e.g., proofs, experimental setup) are sufficiently well-described for competent researchers to confidently reproduce the main results.

**Q3 Main Strengths:**

The key advance in the paper is in an automated framework to construct probabilistic surrogate networks employing recurrent neural net pipelines to approximate stochastic simulators described as probabilistic programs.  The main novelty is in the ability to deal with unbounded number of random variables by decomposing the probability measure over infinite sequences via sequential part decomposition.  The tradeoffs of the method and practical considerations are discussed adequately and the utility of the method demonstrated in specific example s.


**Q4 Main Weakness:**

None that I see.

**Q5 Detailed Comments To The Authors:**

I enjoyed reading the paper.  I am curious about the scalability of the work and its potential applications in vision.

**Q7 Justification For Your Score:**

Overall a solid paper with clear articulation of goals, unique contributions, and illustration of the framework.

**Q9 Complying With Reviewing Instructions:**

1: Yes.

---

### Decision · Program_Chairs · 2022-05-15

**Decision:**

Accept (Poster)

**Comment:**

Meta Review: The paper develops a method for training neural surrogates of stochastic simulators with a focus on retaining the control flow and dealing with simulators where the amount of randomness may be unbounded.

All of the reviewers are supportive of this paper.  The authors engaged in a discussion to clarify concerns from both expert and non-expert reviewers. I strongly suggest the authors incorporate the feedback on clarity based on the questions of reviewer XDYV and the complexity cleanup requested by reviewer n7EN.